# Omics Markers of Red Blood Cell Transfusion in Trauma

**DOI:** 10.3390/ijms232213815

**Published:** 2022-11-10

**Authors:** Ian S. LaCroix, Mitchell Cohen, Ernest E. Moore, Monika Dzieciatkowska, Travis Nemkov, Terry R. Schaid, Margaret Debot, Kenneth Jones, Christopher C. Silliman, Kirk C. Hansen, Angelo D’Alessandro

**Affiliations:** 1Department of Biochemistry and Molecular Genetics, University of Colorado Denver-Anschutz Medical Campus, Aurora, CO 80045, USA; 2Department of Surgery, University of Colorado-Anschutz Medical Campus, Aurora, CO 80045, USA; 3Ernest E Moore Shock Trauma Center at Denver Health, Denver, CO 80204, USA; 4Department of Cell Biology, University of Oklahoma, Oklahoma City, OK 73104, USA; 5Vitalant Research Institute, Denver, CO 80230, USA; 6Department of Pediatrics, University of Colorado-Anschutz Medical Campus, Aurora, CO 80045, USA

**Keywords:** trauma, hemorrhagic shock, plasticizer, succinate, hypoxia, erythrocyte

## Abstract

Red blood cell (RBC) transfusion is a life-saving intervention for millions of trauma patients every year worldwide. While hemoglobin thresholds are clinically driving the need for RBC transfusion, limited information is available with respect to transfusion efficacy at the molecular level in clinically relevant cohorts. Here, we combined plasma metabolomic and proteomic measurements in longitudinal samples (n = 118; up to 13 time points; total samples: 690) from trauma patients enrolled in the control of major bleeding after trauma (COMBAT) study. Samples were collected in the emergency department and at continuous intervals up to 168 h (seven days) post-hospitalization. Statistical analyses were performed to determine omics correlate to transfusions of one, two, three, five, or more packed RBC units. While confounded by the concomitant transfusion of other blood components and other iatrogenic interventions (e.g., surgery), here we report that transfusion of one or more packed RBCs—mostly occurring within the first 4 h from hospitalization in this cohort—results in the increase in circulating levels of additive solution components (e.g., mannitol, phosphate) and decreases in the levels of circulating markers of hypoxia, such as lactate, carboxylic acids (e.g., succinate), sphingosine 1-phosphate, polyamines (especially spermidine), and hypoxanthine metabolites with potential roles in thromboinflammatory modulation after trauma. These correlations were the strongest in patients with the highest new injury severity scores (NISS > 25) and lowest base excess (BE < −10), and the effect observed was proportional to the number of units transfused. We thus show that transfusion of packed RBCs transiently increases the circulating levels of plasticizers—likely leaching from the blood units during refrigerated storage in the blood bank. Changes in the levels of arginine metabolites (especially citrulline to ornithine ratios) are indicative of an effect of transfusion on nitric oxide metabolism, which could potentially contribute to endothelial regulation. RBC transfusion was associated with changes in the circulating levels of coagulation factors, fibrinogen chains, and RBC-proteins. Changes in lysophospholipids and acyl-carnitines were observed upon transfusion, suggestive of an effect on the circulating lipidome—though cell-extrinsic/intrinsic effects and/or the contribution of other blood components cannot be disentangled. By showing a significant decrease in circulating markers of hypoxia, this study provides the first multi-omics characterization of RBC transfusion efficacy in a clinically relevant cohort of trauma patients.

## 1. Introduction

Transfusion of packed red blood cells (pRBCs) is a life-saving intervention for ~4.5 million Americans every year. Patients suffering from trauma and hemorrhagic shock represent one of the major categories of massively transfused recipients. In these patients, transfusion of pRBCs not only contributes to correcting hypovolemia after bleeding, but also and foremost, transfused RBCs restore tissue oxygenation by replenishing the reservoirs of circulating cell hemoglobin. RBC hemoglobin is indeed a trigger for transfusion (10 g/dL is the standard threshold for severely injured patients), and each unit of pRBCs contributes to an increment of ~1 g/dL of hemoglobin. Evidence has also been produced describing the potential impact of RBCs on bleeding and thrombosis as mediated by RBC-derived vesicle and phosphatidylserine exposure on the outer membrane layer of the aged erythrocyte, thereby potentially mediating coagulopathic complications in the severely injured patient [1]. Recent application of omics technologies to the field of transfusion medicine have documented a plethora of biochemical alterations that pRBCs accumulate progressively during refrigerated storage in the blood bank up to 42 days [2]. Despite reassuring evidence from randomized clinical trials (RCTs) [3], recent meta-analyses of the RCTs suggest that transfusing exclusively older RBC units stored for >one or two weeks increases the 28-day recipient mortality and risk of thromboembolism or death compared with transfusing fresher RBC units [4].

The onset, progression, and ultimate severity of the storage lesion are also impacted by donor biological factors, such as sex [5], ethnicity [6], age, body mass index [7], hormone levels [8], or other genetically encoded factors [9]. Similarly, processing strategies, such as the storage additives in blood units [10] or the phthalate plasticizers that leak from them [11], all contribute to the storage quality and molecular heterogeneity of the pRBCs. Altogether, these factors have the potential to influence clinical outcomes in that they impact the propensity of transfused RBCs to hemolyze intra- [12,13] or extra-vascularly [14,15]. In so doing, these factors impact the capacity of storage-damaged RBCs to circulate [16], which in turn limits the efficacy of transfusion therapies, at least with respect to hemoglobin increments [17,18]. In vitro studies have suggested that stored RBCs are also characterized by impaired oxygen kinetics, i.e., the capacity to bind and off-load oxygen [19]. Better preserved oxygen kinetics parameters are observed when RBCs are stored under conditions that mitigate the so-called storage lesion (e.g., anaerobic storage) [20], conditions that improve post-transfusion recovery [21] and facilitate resuscitation in rodent models of trauma and hemorrhagic shock at a fraction of the transfused blood volume [22]. Despite preliminary omics studies on the impact of blood transfusion on the molecular phenotype of transfusion recipients, at least in healthy autologous subjects [23], limited information is known about whether and to what extent pRBC transfusion impacts the circulating metabolome and proteome in trauma patients.

Even after the COVID-19 pandemic, trauma is the leading cause of total life years lost and ranks right after COVID-19 as one of the leading causes of mortality in the United States, accounting for ~5.9% of total yearly deaths in 2020, according to the Center for Disease Control (https://www.cdc.gov/nchs/fastats/leading-causes-of-death.htm, accessed on 5 November 2022). Failure to rescue [24] hypothermia, acidosis, and coagulopathy (hyperfibrinolysis or shutdown) [25] are key drivers of early mortality by trauma [26], while late organ dysfunction and, in particular, acute respiratory distress syndrome appear to be driven by thromboinflammatory events in part triggered by small molecule mediators [27]. While transfusion of pRBCs is a life-saving intervention, studies have been performed to show that some small molecule metabolites that accumulate in stored pRBCs could contribute to triggering inflammatory events (e.g., oxylipins [28]) or coagulopathy by triggering hypocalcemia [29] upon overload with storage additive anticoagulants.

Over the last two decades, omics technologies have contributed significantly to our understanding of the molecular events that are triggered by trauma, hemorrhage, and shock [30,31,32,33,34,35,36]. Proteomics and metabolomics studies have been performed not only in animal models of trauma and shock (e.g., in rodents, swines, or non-human primates) but also in trauma patients [30,31,32,33,34,35,36]. From these studies, molecular markers were linked to mortality, such as lactate and succinate [37], drivers of acidosis following hypoxemia and hypoxia-induced mitochondrial dysfunction [38]. Some of these markers were then mechanistically linked to pro-inflammatory events driving inflammatory complications after trauma [27]. Similar studies were performed to link molecular signatures of trauma to the onset and severity of coagulopathy as determined via correlation of metabolomics [39] and proteomics [40] data and measurements from viscoelastic assays.

## 2. Results

### 2.1. Younger, More Severely Injured, Male Patients Received Most RBC Transfusions in the Emergency Department through 4 h from Hospitalization

Clinical data were obtained for the COMBAT cohort and plotted as a function of packed RBC transfusion events from the emergency department (ED) through the first week (168 h) after hospitalization (Figure 1A). Of the 118 patients in the COMBAT cohort, 61 patients received at least one pRBC transfusion. The transfusions were administered in the ED or at time points between 2 and 24 h post injury, totaling 597 pRBC units. Most of the units were transfused at the ED through the first 4 h post hospitalization (Figure 1A), while the maximum number of pRBCs transfused occurred at 2 h post hospitalization. Conversely, no pRBC units were transfused in the field or after 24 h in this cohort (hence, the focus on this time range in Figure 1). Data were thus broken down by the patients’ clinical (new injury severity scores—NISS and base excess—BE) or biological factors (patients’ age and sex). Based on the former, four main groups were identified: patients with low shock/low injury (n = 34), high shock/low injury (n = 14), low shock/high injury (n = 29), and patients with high shock/high trauma (n = 31—Figure 1B). The number of pRBCs transfused from ED through 24 h were plotted independently by severity phenotype (Figure 1C), patient sex (male (n = 50) and female (n = 11)—Figure 1D), or by age group (Figure 1E). Results show that most transfusion events were recorded in the most severely injured patients (NISS > 25), especially those with severe BE (<−10—Figure 1B). Males were over-represented in these two groups, with the majority of the units being transfused to patients younger than 48. Patients younger than 33 tended to be more heavily transfused not just within 4 h from hospitalization but through the first 12 h interval (Figure 1E).

### 2.2. Metabolic Markers of RBC Transfusion: From Phthalate Plasticizers to Additive Solution Components with an Emphasis on Circulating Markers of Hypoxia

We then set out to determine omics correlate to pRBC transfusion in the plasma of trauma patients. First, we focused on small molecule metabolites by performing Spearman correlations of the total RBC units with all detected metabolites (Figure 2). Given the heterogeneous distribution of RBC transfusion as a function of NISS and BE, correlations were performed after breaking down the cohort as a function of severity groups (Figure 2A). Since most of the units were transfused at the ED at the 2 h and 4 h time points, these time points were used for the correlations. Temporal trends were thus plotted as line plots based on longitudinal data for the top significant metabolite correlates to RBC transfusion across the four groups (Figure 2B). As part of the same analysis and as an internal reference for control values, data from trauma patients were compared against the same measurements in plasma from healthy volunteer (HVs). Line plots were thus color-coded (shades of blue) to mirror the number of RBC units received (0, 1, 2, or 3–5 packed RBC transfusions—Figure 2B) at each time point from ED to 168 h (one week post hospitalization) per each group. Expectedly, the top correlates to packed RBC transfusion events included additive solution components (mannitol, di- and ortho-phosphate) or metabolites previously reported [10] to be strongly associated with the metabolic impact of additive solution on stored RBC metabolism (e.g., S-adenosyl-homocysteine—Figure 2B). In the groups with the highest number of transfusions (NISS > 25), the top correlates also included metabolic markers of the RBC storage lesion (e.g., hypoxanthine) [41] as well as metabolic products of heme catabolism (biliverdin and bilirubin—Figure 2B) and polyamines spermidine and spermine (Figure 2B).

### 2.3. Plasma Metabolic Correlates to Packed RBC Transfusion as a Function of the Number of Units Transfused

After this preliminary analysis, we thus broke down the cohort as a function of transfusion events (how many pRBCs were transfused) to determine whether dose effect responses were notable on the circulating metabolome of transfusion recipients. First, we mapped out the number of transfusions received during the time window monitored in this longitudinal study (from ED to 1 week post hospitalization—Figure 3). Multiple heat maps were thus generated to identify changes in the plasma metabolome at the time points preceding (pre), during, and after (post) the transfusion event (Figure 4) as a function of the number of units transfused (1 to 5—Figure 4A). Partial least square-discriminant analysis (PLS-DA) on patients that received two or more transfusions showed that this classification (pre, during, post transfusion) was sufficient to cluster the samples, though it explained only 5.1% of the total variance (Figure 4B). However, these combined analyses were instrumental to the identification of metabolites whose levels increased/decreased in the plasma of transfusion recipients as a function of the dose (number) of transfused pRBC units (Figure 4C). Indeed, we observed decreases in the levels of markers of hypoxia (hypoxanthine, sphingosine 1-phosphate, malate), arginine catabolites involved in inflammation and endothelial modulation (ornithine and citrulline), oxylipin products of inflammatory and oxidation events (12-HETE), and lysophospholipids (specifically, lysophosphatidylcholines—LPC 14:0; and lysophosphatidylethanolamines—LPE 18:1 and 16:0—Figure 4C).

Expectedly, transfusion events were positively associated with the accumulation of phthalate plasticizers (mono-ethyl-hexyl- and di-ethyl-hexyl-phthalate—MEHP and DEHP) in the plasma of transfusion recipients (Figure 5A). DEHP levels in plasma (ED and 2 h time points) were significantly higher in the patients receiving a higher number of pRBC units, especially early upon transfusion (ED through 4 h time points—Figure 5B). Metabolic correlates to plasma DEHP levels in transfusion recipients (Figure 5C) almost entirely overlapped with the top metabolic correlates to the number of packed RBC transfusion events (Figure 2). Higher DEHP levels were associated with a higher need for transfusion (i.e., higher levels of circulating markers of hypoxia, succinate, and lactate—Figure 5C). The top significant correlates were fit by a simple linear model; the *p*-value and R squared of the slope are reported above each plot (Figure 5D).

### 2.4. Protein Correlates to Packed RBC Transfusion in the Plasma of Trauma Patients

Similar analyses were performed to correlate plasma protein levels to pRBC transfusion events in the same cohort (Figure 6A). Independent Spearman correlation analyses were performed based on severity groups (same as above), as indicated by BE and NISS cut-off values (−10 and 25, respectively). In all four groups, the top positive correlates to transfusion events included multiple RBC proteins, such as the most abundant cytosolic (hemoglobins—HBA and HBB; biliverdin reductase B—BLVRB; peroxiredoxin 2—PRDX2) and membrane proteins (anion transporter band 3 (B3AT), stomatin (STOM), carbonic anhydrase (CAH1)—Figure 6A). Temporal trends were thus plotted for the most significant protein correlates identified, as plotted against the mean values measured for the same analytes in healthy volunteers (HV—Figure 6B). In these plots, the patients were grouped by number of RBC units received (0, 1, 2, or 3–5), showing a dose-response effect in plasma levels of gelsolin (GELS—Figure 6B). To further expand on the dose-response observation, we broke down the cohort by the number of pRBC units transfused and generated heat maps by dividing protein levels into transfusion phases “pre-during-post” (Figure 7A). PLS-DA on proteomes of patients that received two or more transfusions grouped by transfusion phase, revealed significant increases/decreases in the levels of a subset of plasma proteins as a function of RBC transfusion events (dose response—Figure 7B). Increasing proteins included several coagulation components (FIBA and FIBB) and factor (FA9), metalloproteases (TIMP1), complement component (CO7), lipopolysaccharide binding protein (LBP), apolipoprotein E (APOE), and ceruloplasmin (CERU). Opposite trends were observed for BLVRB (Figure 7C).

### 2.5. Integrated Multi-Omics Pathway Analysis of Molecular Correlates to pRBC Transfusion

To integrate the results described above, we performed pathway analyses of significantly positive or negative correlates to pRBC transfusion in the plasma metabolome or proteome of COMBAT patients (Figure 8A,B). Results indicate a clear positive regulation of hydrogen peroxide and aldehyde detoxification processes, especially those that are dependent on glutathione and NADPH as a reducing equivalent (e.g., glutathione peroxidases, glutathione s-transferases, peroxiredoxins, aldehyde dehydrogenases) and other antioxidant pathways (S-adenosylmethionine metabolism, superoxide dismutases—Figure 8C). Of note, the third most up-regulated pathway was involved in oxygen transport and sequestering of iron, consistent with the main purpose of the pRBC transfusion. Several metabolic proteins involved in metabolic processes were observed in this network (lactate dehydrogenases, enolase, phosphoglucomutase—Figure 8A,C,E). With respect to negatively regulated processes following pRBC transfusion, pathway analyses pointed at a decrease in the levels of molecules involved in the negative regulation of hydrolase, peptidase, and proteolytic activity, especially (serine) protease inhibitors that participate in coagulation responses and wound healing (Figure 8B,D,E).

Transfusion of pRBCs, perhaps in part influenced by the concomitant transfusion of other blood components, was also associated with decreases in protein and metabolic markers of acute phase response, inflammatory responses, and viral catalytic/molecular function (Figure 8D).

## 3. Discussion

Omics applications in the fields of emergency and transfusion medicine have fueled new discoveries leading to improved understanding of hemorrhagic hypoxia [30,31,32,33,34,35,36] and the RBC storage lesion to the extent storage quality [2] is affected. However, limited data and tools to generate them are available to evaluate the efficacy of the transfused RBCs in vivo, as highlighted by a recent NIH-sponsored workshop [42]. As part of this workshop, experts in the fields of transfusion and emergency medicine, biochemistry, and omics studies shared the “latest findings, discussed key challenges, and identified opportunities for facilitating development of new technologies and/or biomarker panels to assess tissue oxygenation in a minimally-invasive to noninvasive fashion before and after RBC transfusion” [42]. The present study directly responds to the NIH call for bridging this knowledge gap by providing a direct characterization of the molecular signatures of pRBC transfusion in the plasma of a real-world cohort of trauma patients. Here, we combined a careful longitudinal sampling of clinically characterized trauma patients as stratified by NISS and BE with state-of-the-art metabolomics and proteomics (paired) analyses of hundreds of samples. Results provide evidence of a direct association (and a dose effect) between changes in recipients’ plasma metabolites and protein levels and pRBC transfusion, especially in the critically ill, young male patients enrolled in this study and mostly transfused within 4 h from hospitalization. Specifically, transfusion of pRBCs was associated with decreases in the circulating levels of metabolic markers of hypoxia. Examples include decreases in carboxylic acids, such as malate and succinate. Most notably, subjects receiving the highest number of transfusions were proportionally more likely to have higher initial levels of succinate, an association that in this cohort did not hold necessarily true for lactate, another clinical marker of severity of trauma and shock. This result is suggestive that early correction of circulating levels of succinate and lactate is a critical hallmark of resuscitation, suggesting that succinate is not only a marker of mortality [37] but also a key marker of systemic hypoxemia and transfusion efficacy in trauma patients. Succinate is an etiological contributor to acute inflammatory distress syndrome since this dicarboxylate can trigger succinate receptor 1 and thus promote neutrophil infiltration in the lung [27]. Decreases in the levels of sphingosine 1-phosphate were observed upon pRBC transfusion. Sphingosine 1-phosphate is another marker of hypoxia which is mostly derived from RBCs [43] and can trigger T cell extravasation from the lymphatic [44]. Similarly, pRBC transfusion decreased the levels of circulating hypoxanthine. Hypoxanthine, a product of high-energy purine breakdown and deamination in response to ischemic or hemorrhagic hypoxia [38], can serve as a substrate for the synthesis of urate, a reaction that concomitantly releases pro-oxidant hydrogen peroxide. Polyamines, such as spermidine [31,32,34], have the potential to activate phagocytic cells and mediate efferocytosis [45] of necrotic cells upon tissue damage. Mitochondrial dysfunction in the face of hypoxia as well as elevation in the circulating levels of free iron (e.g., following transfusion-associated hemolysis [46] or intravenous injection of iron chloride [47]) all impact circulating levels of polyamines, shown here to be impacted by pRBC transfusion.

Small molecule metabolites derived from additive solutions [10] (e.g., mannitol, phosphates) or phthalate plasticizers [48] were observed to transiently increase in the bloodstream of the recipient. One limitation of the present study is that no information was recorded/accessible on the age of the transfused units nor on the additive in which pRBCs were stored. However, high levels of mannitol are suggestive of non-mannitol-free additives being used here (e.g., AS-1 and AS-5), which in part explains why no significant elevation of citrate was noted upon transfusion. As such, future studies on other mannitol-free, citrate-loaded additives (e.g., AS-3) may provide clues on whether pRBC transfusion can trigger hypocalcemia in trauma patients, which has been recently questioned as a contributor to coagulopathy of trauma [29,49]. Similarly, since phthalates leach from plastic bags as a function of storage duration [48], it is interesting to note that phthalate levels increased non-linearly with the number of transfused units, suggestive of an increased likelihood of older units being transfused to more severely injured patients requiring larger volumes of transfusion. This observation may hold clinical implications in that phthalate plasticizers, which can reach millimolar levels by the end of the shelf-life of the unit, are known to exert cardioactive effects and may thus interfere with resuscitation and stabilization of the patient [50].

The present study holds several limitations. First, these patients were not only receiving pRBC transfusion but were also concomitantly undergoing (more or less) invasive surgery other than being transfused with plasma and platelet products which may have impacted our findings. Reassuringly, oxygen transport and antioxidant metabolic pathways critical processes regulated by RBCs at the systemic level were the top positively correlated pathways to pRBC transfusion, as gleaned from integrated omics analyses. Notably, similar omics correlates to transfusion events were observed after stratification of patients based on NISS and BE, suggestive of constant molecular signatures of pRBC transfusion independent from mode and severity of injury. Reassuringly, the observed increases in circulating levels of RBC-derived proteins upon transfusion are also suggestive that at least in part the markers of pRBC transfusion reported herein are at least in part real signatures. On the other hand, transfusion of pRBCs was here associated (in a dose-response fashion) to increases in the circulating levels of several coagulation cascade components, complement components, and fibrinogen components, suggestive of (i) a previously unappreciated impact of RBC transfusion on these pathways; (ii) a (more likely) concomitant effect of platelet and plasma product transfusion on the circulating proteome. In the present study, we have no information on the characteristic of pRBC units transfused here, including storage age, donor biology (including sex, age, body mass index, genetics), or other exposures, that may have impacted RBC storage biology and potentially, transfusion efficacy as these factors do indeed associate with hemoglobin increments [18] and molecular signatures in the plasma of transfusion recipients. Despite the many limitations, our study paves the way for future explorations on the impact of pRBC transfusion on the circulating molecular make up of transfusion recipients, such as the ongoing recipient epidemiology and donor evaluation study [51].

## 4. Materials and Methods

### 4.1. Ethics Statement

Patients were enrolled under a waiver of informed consent, as permitted under U.S. Federal Regulation 21 CFR 50.24. To comply with this regulation, the study was conducted after a process of community consent, wherein the local community was informed via multiple media outlets and was provided with the option to opt out by wearing a “No COMBAT” bracelet or dog tags. Additionally, the waiver of consent element of the study design required that the study was conducted under FDA regulation as an investigational new drug study (IND no. 15216).

### 4.2. Patient Enrollment

An overview of the COMBAT study was provided in previous reports [52,53]. Briefly, severely injured patients were enrolled at the Denver Health Medical Center, a level I Trauma Center in Denver, Colorado (inclusion criteria: age > 18; acutely injured; SBP < 70 mm Hg or SBP 71–90 mm Hg with heart rate > 108 beat per minute; exclusion criteria: visibly or verbally reported pregnant women, known prisoners, unsalvageable injuries defined as asystolic or cardiopulmonary resuscitation prior to randomization; known objection to blood products; patients with opt-out bracelet, necklace, or wallet card; family member present at the scene objecting to patient’s enrollment in research). A total of 690 longitudinal samples (n = 118) were collected either in the field (<30 min from traumatic injury), at the arrival in the emergency department (ED), or at 2, 4, 6, 12, 24, 48, or 72 h or five or seven days from the injury. Thirteen patients died in the ED or within two hours from the injury or within the first 48 h. Whole blood samples were collected in EDTA and immediately centrifuged at 2500× *g* for 10 min at 4 °C (in the ambulance, ED, or hospital) to sort plasma from blood components prior to storage at −80 °C and subsequent extraction for metabolomics analyses.

### 4.3. Ultra-High-Pressure Liquid Chromatography-Mass Spectrometry (MS) Metabolomics

Frozen plasma aliquots (10 µL) were extracted 1:25 in an ice cold extraction solution (methanol:acetonitrile:water 5:3:2 *v*/*v*/*v*). Samples were vortexed for 30 min at 4 °C prior to centrifugation for 10 min at 15,000× *g* at 4 °C, as described [54,55]. Analyses were performed using a Vanquish UHPLC coupled online to a Q Exactive mass spectrometer (Thermo Fisher, Bremen, Germany). Samples were analyzed using a 1 min and 5 min gradient-based method, as described [54,55].

### 4.4. Proteomics Analyses via Nano-UHPLC-MS/MS

Plasma samples were digested in the S-Trap 96-well plate (Protifi, Huntington, NY, USA) following the manufacturer’s procedure. Briefly, around 50 μg of plasma proteins were first mixed with 5% SDS. Samples were reduced with 10 mM DTT at 55 °C for 30 min, cooled to room temperature, and then alkylated with 25 mM iodoacetamide in the dark for 30 min. Protein thiols were reduced with 10 mM DTT at 55 °C for 30 min, followed by cooling to room temperature and then alkylation with 25 mM iodoacetamide in the dark for 30 min. Next, a final concentration of 1.2% phosphoric acid and then six volumes of binding buffer (90% methanol; 100 mM triethylammonium bicarbonate, TEAB; pH 7.1) were added to each sample. After gentle mixing, the protein solution was loaded to a S-Trap 96-well plate, spun at 1500× *g* for 2 min, and the flow-through collected and reloaded onto the 96-well plate. This step was repeated three times, and then, the 96-well plate was washed with 200 μL of binding buffer three times. Finally, 1 μg of sequencing-grade trypsin (Promega) and 125 μL of digestion buffer (50 mM TEAB) were added onto the filter and were digested out at 37 °C for 6 h. To elute peptides, three stepwise buffers were applied with 200 μL of each with one more repeat, including 50 mM TEAB, 0.2% formic acid in H_2_O, and 50% acetonitrile and 0.2% formic acid in H_2_O. The peptide solutions were pooled, lyophilized, and resuspended in 500 μL of 0.1% FA. Those samples were diluted further 10-fold with 0.1% FA. A 20 μL aliquot of each sample was loaded onto individual Evotips for desalting and then washed three times with 200 μL 0.1% FA followed by the addition of 100 μL storage solvent (0.1% FA) to keep the Evotips wet until analysis. The Evosep One system (Evosep, Odense, Denmark) was used to separate peptides on a Pepsep column (150 µm inter diameter, 15 cm) packed with ReproSil C18 1.9 um, 120 A resin. The system was coupled to the timsTOF Pro mass spectrometer (Bruker Daltonics, Bremen, Germany) via the nano-electrospray ion source (Captive Spray, Bruker Daltonics). The mass spectrometer was operated in diaPASEF mode. We used a method with four windows in each 100 ms dia-PASEF scan. Thirty-two of these scans covered the diagonal scan line for doubly and triply charged in the *m/z*-ion mobility plane with narrow 25 *m/z* precursor windows. A project-specific library generated from 24 high-pH reverse-phase peptide fractions acquired with PASEF technology. MS data were collected over an *m/z* range of 100 to 1700. During each MS/MS data collection, each TIMS cycle was 1.17 s and included 1 MS and 10 PASEF MS/MS scans. Low-abundance precursor ions with an intensity above a threshold of 500 counts but below a target value of 20,000 counts were repeatedly scheduled and otherwise dynamically excluded for 0.4 min. Pooled plasma digest was separated by high pH reversed phase chromatography on a Gemini-NH C18, 4.6 × 250 mm analytical column containing 3 μM particles; flow rate was 0.6 mL/min. The solvent consisted of 20 mM ammonium bicarbonate (pH 10) as mobile phase A and 20 mM ammonium bicarbonate and 75% ACN (pH 10) as mobile phase B. Sample separation was accomplished using the following linear gradient: from 0 to 5% B in 10 min, from 5 to 50% B in 80 min, from 50 to 100% B in 10 min, and held at 100% B for an additional 10 min. A total of 96 fractions were collected along with the LC separation and were concatenated into 24 fractions by combining fractions 1, 25, 49, 73, and so on. Samples were dried in Speed-Vac, resuspended in 80 μL of 0.1% FA, and 20 μL of each fraction was loaded onto individual Evotips.

### 4.5. Data Analysis

Spearman correlations were performed in R using total RBC units received against all metabolites or proteins, run independently for each one of the four groups based on new injury severity scores (NISS—thresholds 25) and base excess (threshold 10) at the emergency department (ED) time point. A multiple linear regression model, as previously described, was used to impute BE for 20 patients with missing values at the ED time point. Using the BE and NISS, patients were categorized into four injury severity groups. Intensity levels of the top metabolite or protein correlates were plotted over the complete time course. Groups indicate the total number of RBC units received. The in-house developed code in R (version 4.1.0; 18 May 2021) was used to extract patients that received only one transfusion during the time course. The number of RBC units received during the single transfusion ranged from one to five, depending on electronic medical records. The metabolomics and proteomics profiles of these patients were analyzed using MetaboAnalyst 5.0 [56], and the top 25 ANOVA analytes were visualized by heat map “pre”, “during”, and “post” transfusions. The “pre”, “during”, “post” time points were dynamically defined per patient receiving one transfusion (n = 20). “Pre” included all time points leading up to the single transfusion. For this analysis, “during” consisted of the single transfusion time point. While “post” included all time points after the transfusion. Analytes that showed levels that were significantly different (by *t*-test) “pre” compared to “post” were visualized by violin plots. Patients that received multiple transfusion events (n = 42, not considering number of RBC units) were analyzed separately. These patients were further grouped by the specific number of transfusion events which ranged from two to five. The top 25 ANOVA analytes were visualized by heatmaps using MetaboAnalyst. The Omicsnet software package (https://www.omicsnet.ca/, accessed on 5 November 2022) was used for integrated network and pathway analyses. The “pre”, “during”, and “post” time points were dynamically defined per patient receiving multiple transfusion events. For this portion of the analysis, “during” was defined as any inclusive time point between the first and last transfusion event. Not all patients received transfusions at consecutive time points. This loose definition of the “during” time point enriched the number of patients included in each transfusion event group.

## Figures and Tables

**Figure 1 ijms-23-13815-f001:**
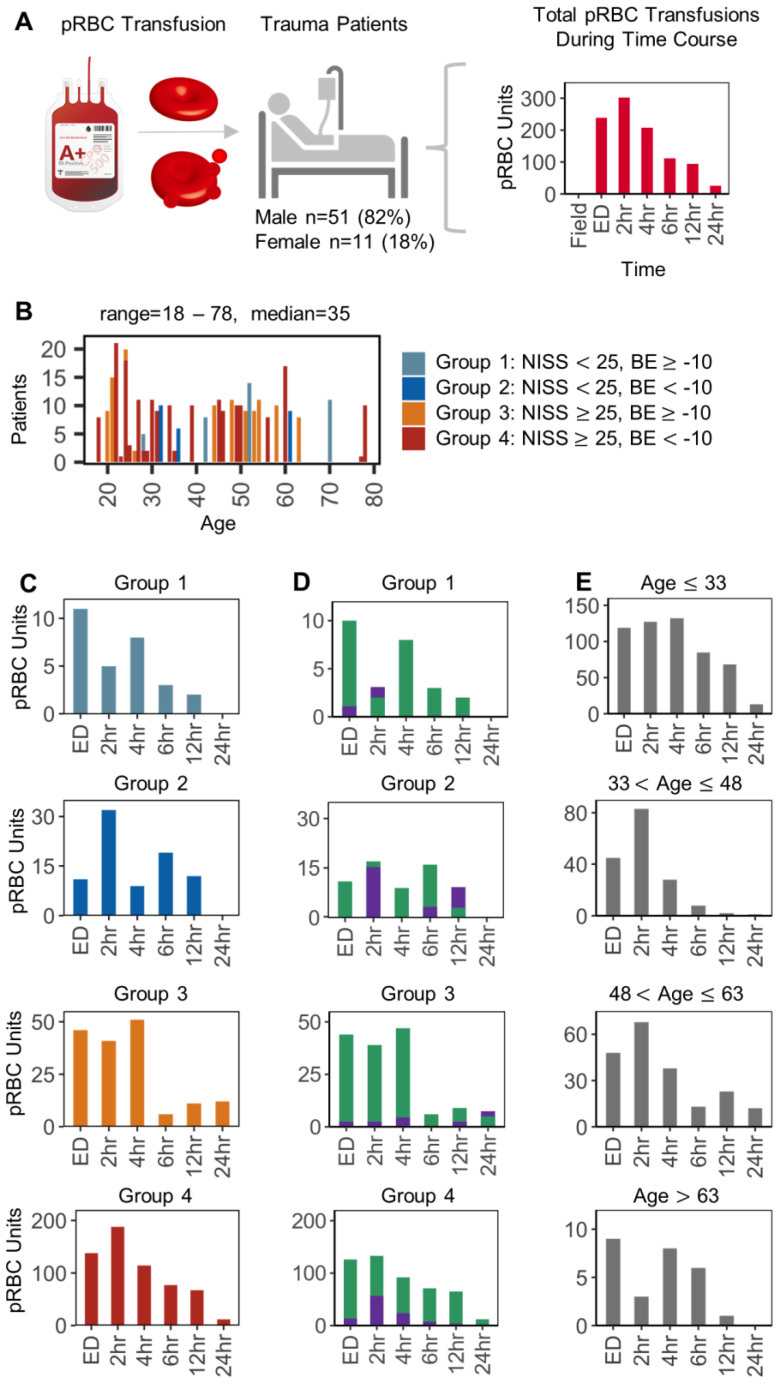
Overview of RBC transfusions in the COMBAT patient cohort. In (**A**), an overview of the experimental design and time distribution of pRBC transfusion events. In (**B**), subject age distribution of pRBC transfusion upon identification of four groups based on new injury severity score (NISS) and base excess (BE). Bar colors differentiated severity phenotype with light blue: Group 1 (n = 10), dark blue: Group 2 (n = 8), orange: Group 3 (n = 16), red: Group 4 (n = 27). The number of pRBC units transfused from ED-24 h were plotted independently by severity phenotype (**C**), sex (**D**): male (green, n = 50) and female (purple, n = 11), or by age (**E**).

**Figure 2 ijms-23-13815-f002:**
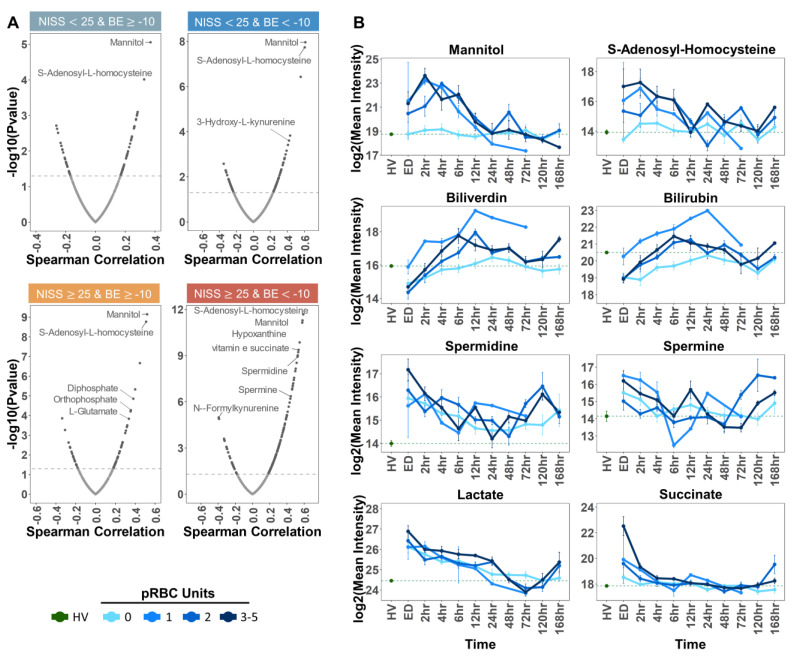
Metabolite correlates to pRBC transfusion Spearman correlations of total pRBC units with all detected metabolites (**A**). Correlations were performed per severity group as indicated by BE and NISS cut-off values labeled above each plot. Patient samples at the ED time point were used for the correlations. The Spearman correlation values were plotted on the *x*-axis, and −log10 transformed *p* values were plotted on the *y*-axis. The dashed horizontal line indicates the significance threshold, −log10(0.05). Temporal trends of top significant metabolite correlates are shown in panel (**B**). The dashed green line indicates the log2 transformed mean healthy volunteer (HV) intensity. Patients were grouped by number of pRBC units received (0, 1, 2, or 3–5). The log2 transformed mean patient intensity was plotted at each time point from ED to 168 h per group. Bars represent the standard error at each time point.

**Figure 3 ijms-23-13815-f003:**
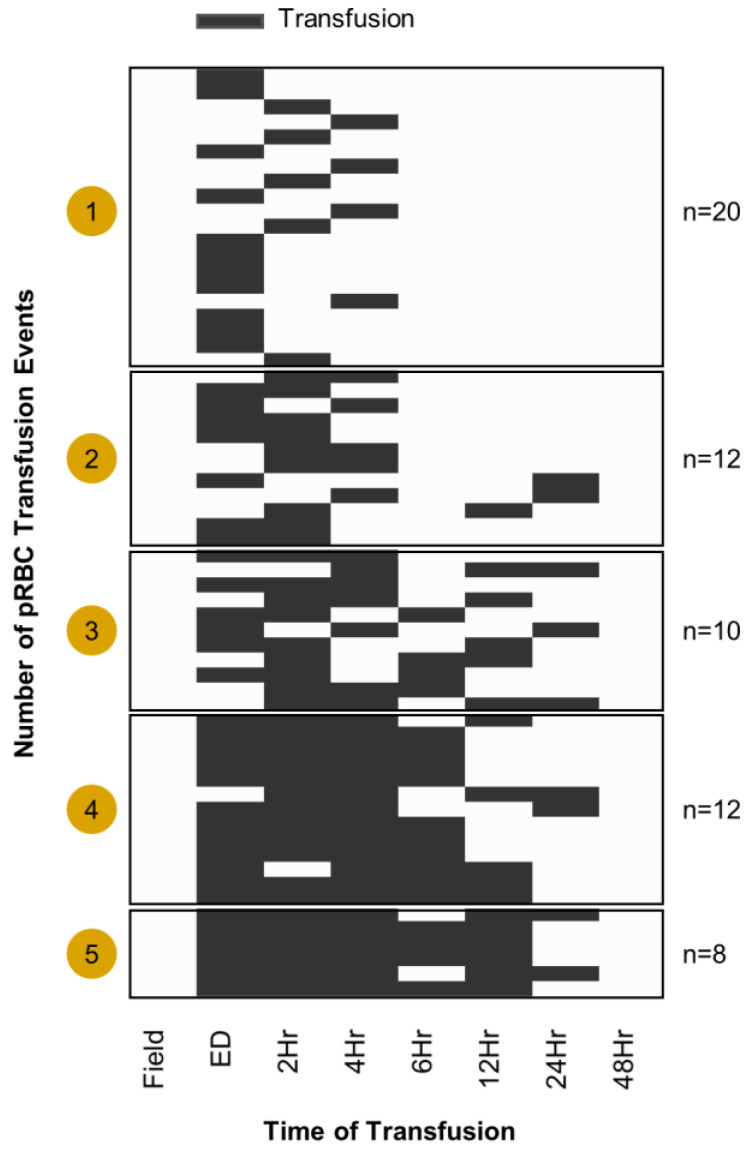
Heat map of patients that received transfusions during the time course. Black coloring represents when a transfusion took place. Patients were grouped by the number of transfusion events where “events” ranged from one to five transfusions.

**Figure 4 ijms-23-13815-f004:**
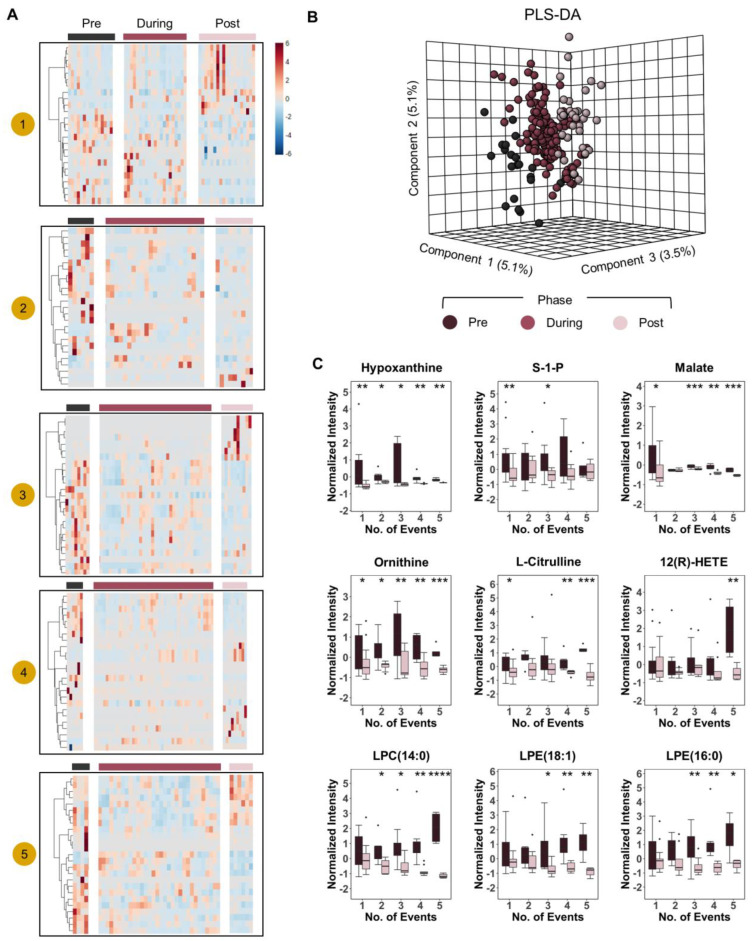
Dose-response effect of pRBC transfusion on the plasma metabolome heat maps of top 25 metabolites that changed significantly in response to transfusion as measured by ANOVA (*y*-axis) with patients on the *x*-axis (**A**). The heat maps were divided into transfusion phases “pre-during-post” grouped by the number of transfusion “events” 1–5. In panel (**B**), PLS-DA of patient metabolomes grouped by transfusion phase. Patients that received strictly greater than one transfusion event were included in PLS-DA. In (**C**) we show box plots of select metabolites that changed significantly in response to transfusions. Box plot colors correspond to pre and post transfusion time points, consistent with panels (**A**,**B**). **** *p* < 0.0001, *** *p* < 0.001, ** *p* < 0.01, * *p* < 0.05.

**Figure 5 ijms-23-13815-f005:**
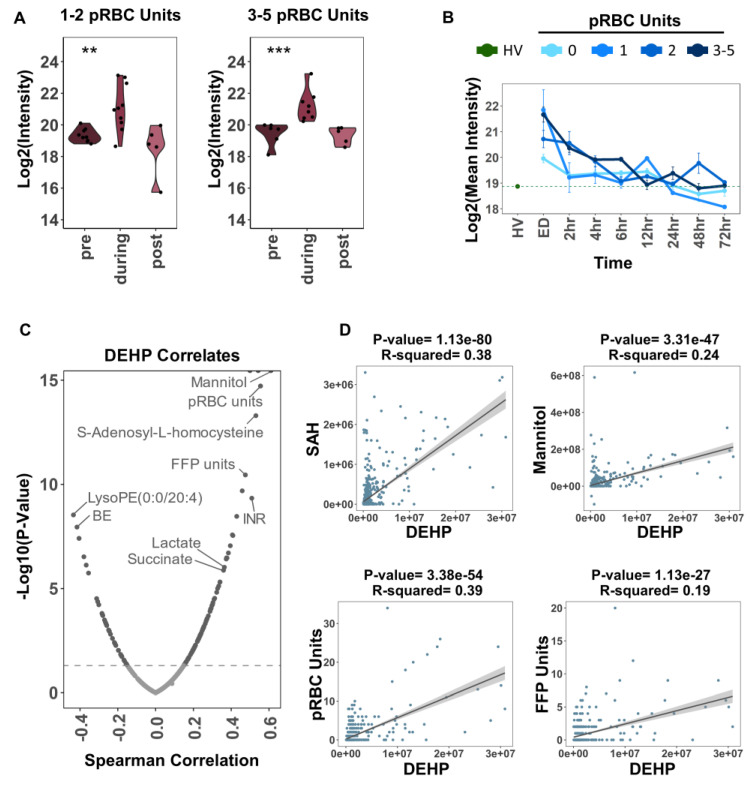
Impact of pRBC transfusion on circulating levels of common plasticizers that leach from the blood bag during storage of erythrocyte concentrates. In (**A**), Log2 transformed DEHP intensities pre, during, and post transfusion. For patients that received 1–2 or 3–5 pRBC units during transfusion. Significance between phases were measured by ANOVA. *** *p* < 0.001, ** *p* < 0.01. In (**B**), Log2 transformed mean DEHP intensity for healthy volunteers compared to trauma patient temporal trends separated by the number of pRBC units received during transfusion. Error bars represent the standard error at each time point. In (**C**), DEHP for trauma patient time point samples from ED and 2 h were correlated (Spearman method) with all metabolites and selected clinical labs: base excess (BE), INR, FFP units, pRBC units. In (**D**), selected significant positive correlates (as determined in panel (**C**)) were fit by a simple linear model. The 95% CI of the slope is shaded in grey surrounding the regression line. The *p*-value and R squared of the slope are reported above each plot.

**Figure 6 ijms-23-13815-f006:**
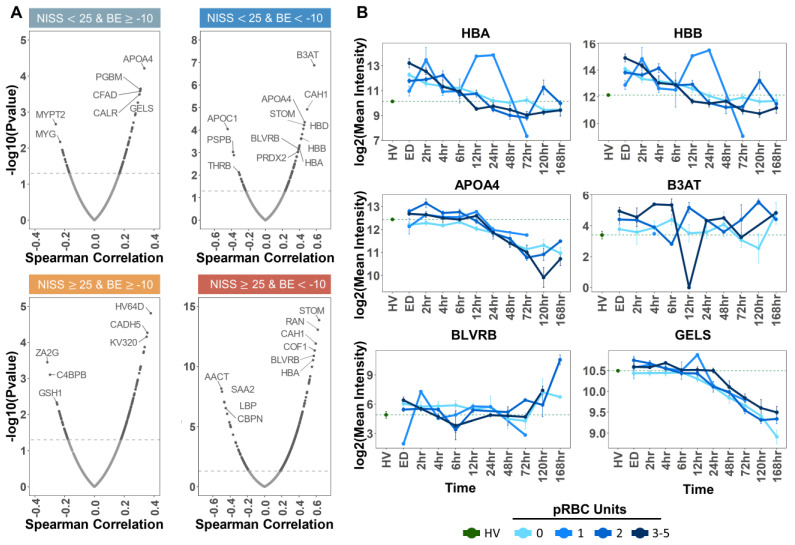
Protein correlates to RBC transfusion. In (**A**), Spearman correlations of total pRBC units with all detected proteins. Correlations were performed per severity group as indicated by BE and NISS cut-off values labeled above each plot. Patient samples at the ED time point were used for the correlations. The Spearman correlation values were plotted on the *x*-axis, and −log10 transformed *p* values were plotted on the *y*-axis. The dashed horizontal line indicates the significance threshold, −log10(0.05). In panel (**B**), temporal trends of top significant protein correlates identified in (**A**). The dashed green line indicates the log2 transformed mean healthy volunteer (HV) intensity. Patients were grouped by number of pRBC units received (0, 1, 2, or 3–5). The log2 transformed mean patient intensity was plotted at each time point from ED to 168 h per group. Bars represent the standard error at each time point.

**Figure 7 ijms-23-13815-f007:**
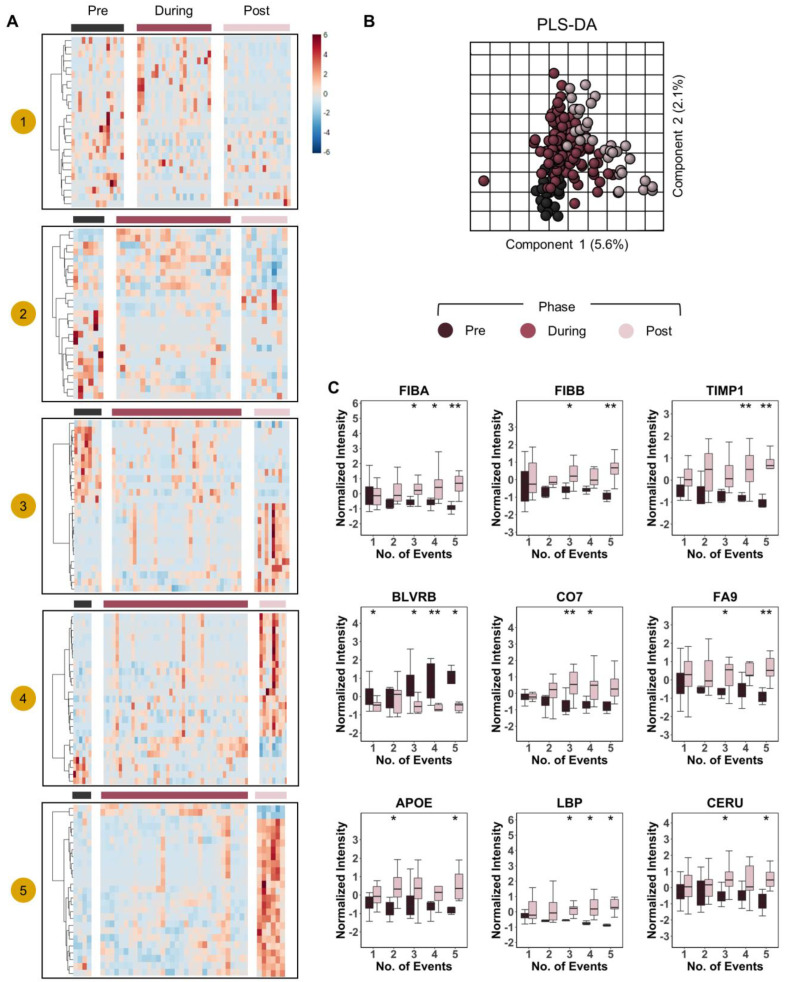
Dose-response effect of pRBC transfusion on the plasma proteome. In (**A**), heat maps of top 25 proteins that changed significantly in response to transfusion as measured by ANOVA (*y*-axis) with patients on the *x*-axis. The heat maps were divided into transfusion phases “pre-during-post” and grouped by the number of transfusion “events” 1–5. In (**B**), PLS-DA of patient proteomes grouped by transfusion phase. Patients that received strictly greater than one transfusion event were included in PLS-DA. In (**C**), we show box plots of select proteins that changed significantly in response to transfusion. Box plot colors correspond to pre and post transfusion time points, consistent with panels (**A**,**B**). ** *p* < 0.01, * *p* < 0.05.

**Figure 8 ijms-23-13815-f008:**
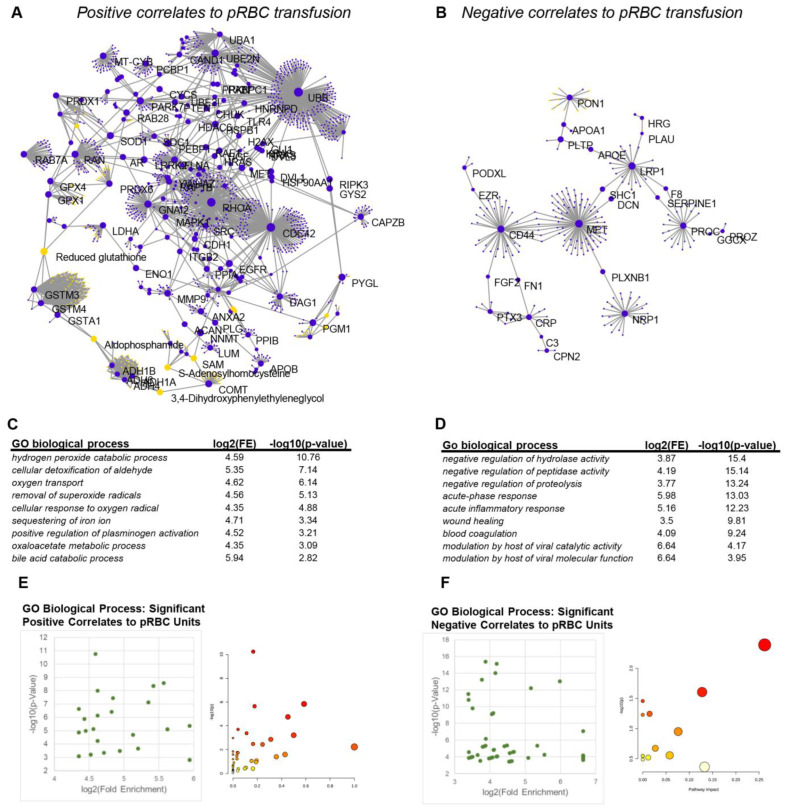
Integrated pathway analysis of metabolomics and proteomics data identifies pathways that are positively and negatively impacted by pRBC transfusion. In (**A**,**B**), network analysis of combined metabolomics and proteomics data that positively or negatively correlate to pRBC transfusion, respectively. In (**C**,**D**), pathway analyses of the significant positive and negative correlates to pRBCs in COMBAT. In (**E**,**F**), score plots of the pathways enriched in the analyses in (**C**,**D**).

## Data Availability

The raw data associated with this study are available from the corresponding author upon reasonable request.

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
