# Peer review of "Omics Markers of Red Blood Cell Transfusion in Trauma"

_ijms, 2022, doi:10.3390/ijms232213815_

Round 1

Reviewer 1 Report

Lacroix et al present an outstanding set of metabolomic and proteomic data from plasma specimens derived from longitudinally (7 days) followed trauma subjects who were transfused within the randomized COMBAT trial (Lancet 2018). The manuscript is extremely interesting because it provides correlative information on the effects of transfusion on specific metabolites and proteins in plasma. The investigators data do support that the transfusion of one or more packed RBCs results in the early increase in circulating levels of additive solution components and decreases in the levels of circulating markers of hypoxia and arginine metabolites, especially in the sickest patients, and the effect observed correlates with the number of units transfused. RBC transfusion was associated with changes in the circulating levels of coagulation factors, fibrinogen, and RBC derived proteins. Also, they are able to detect transient increases the circulating levels of leached plasticizers.

The manuscript is very well written and is comprehensive. Data interpretation is limited by the large number of confounders in the context of a complex trial in trauma, but the authors are careful in not overinterpreting the data and present their correlative descriptive data without mechanistic interpretations. Their discussion carefully points out to the limitations of the study.

Legends for Figures 4, 5 and 7 would benefit from inclusion of a title.

Author Response

REVIEWER 1 Lacroix et al present an outstanding set of metabolomic and proteomic data from plasma specimens derived from longitudinally (7 days) followed trauma subjects who were transfused within the randomized COMBAT trial (Lancet 2018). The manuscript is extremely interesting because it provides correlative information on the effects of transfusion on specific metabolites and proteins in plasma. The investigators data do support that the transfusion of one or more packed RBCs results in the early increase in circulating levels of additive solution components and decreases in the levels of circulating markers of hypoxia and arginine metabolites, especially in the sickest patients, and the effect observed correlates with the number of units transfused. RBC transfusion was associated with changes in the circulating levels of coagulation factors, fibrinogen, and RBC derived proteins. Also, they are able to detect transient increases the circulating levels of leached plasticizers. The manuscript is very well written and is comprehensive. Data interpretation is limited by the large number of confounders in the context of a complex trial in trauma, but the authors are careful in not overinterpreting the data and present their correlative descriptive data without mechanistic interpretations. Their discussion carefully points out to the limitations of the study.

 Authors’ reply: Thank you for the detailed and enthusiastic summary of our manuscript!

Reviewer’s comment: Legends for Figures 4, 5 and 7 would benefit from inclusion of a title.

Authors’ reply: A title was added to figure legends 4, 5 and 7.

Reviewer 2 Report

Comments to the Authors

General comments:

In this work an untargeted metabolomics and proteomics analysis was performed on plasma from trauma patients also divided into groups based on levels of Shock and Injury. I believe the study is interesting and innovative, but here I suggest some points the Authors could find interesting to discuss on:

Major points:

-        -The authors state in the abstract and throughout the text that they performed a combined metabolomics and proteomics analysis, but, in my opinion, the metabolomics and proteomics data are not integrated. Therefore it is advisable to try to integrate the omics data or otherwise remove the word "combined";

-       - In the Results section 2.1, as well as in the abstract, 95 patients are cited, but counting the patients of the four subgroups the patients seem to be 108, the casuistry used should be better explained;

-        -Figure 1F is described in the same section, but it is not located between the panels of Figure 1;

-        -The temporal trends present in Figures 2, 5, and 6 are difficult to read as the shades of blue colors are difficult to correlate to the reference pRBC Units; I recommend using 5 more diversified colors;

-       - The description in Figure 1 seems very close to a paragraph of results; the caption should be reformulated by making only the description of the panels in the figure;

-        -In 2.2 Paragraph Figure 1 is cited instead to Figure 2, please check carefully;

-       -The discussion of the data seems to be very unbalanced in the context of metabolomics and not very thorough with respect to the data obtained from proteomics. An enrichment analysis or a  pathways analysis of proteomics data or integrated ones (metabolites and proteins) may help the reading the obtained results in a functional point of view.

Minor points:

-        In the Introduction (lines 97-99) the sentence could be simplified for greater clarity in the reading phase.

-        In the captions of the figures, punctuation could be used to separate the description of each panel that make up the figure (example: “A) pRBC transfusions […]” - line 154).

-        In the Results section 2.3 there is probably a typo in the phrase “This group included (decreases in) circulating markers of […]” - line 190.

-        In the Figure 6 of the Results section 2.4 probably the point is missing from the generic description of the figure (line 253) and in line 258 probably the authors wanted to indicate “protein” instead of “metabolite”.

-        In the Figure 7 of the same section there is probably a typo in the phrase “Box plots of select proteins […]” - line 268.

-        In the Discussion section there is probably a typo in the phrase “was associated with decreases the circulating […]” - line 290 - and at line 294 the word “and” is missing before “lactate”.

-        In the same section al line 327 maybe the “,” should be replaced with “-“ as in the previous line 326.

-        In the Materials and Methods section 4.2 gaps may be left as in the following example: SBP<70mmHg - line 364, 369.

-        In the Materials and Methods section 4.4 the authors could double check the syntax of the sentence at line 385 and the unit of measure described at lines 401, 405, 425. Moreover, there are some extra spaces at lines 388, 406, 411. At the end, the point is missing at the end of the sentence at line 426.

-        In the Materials and Methods section 4.5 the words “pre”, “post” and “during” should be written in quotation marks to facilitate reading.

Author Response

REVIEWER 2 In this work an untargeted metabolomics and proteomics analysis was performed on plasma from trauma patients also divided into groups based on levels of Shock and Injury. I believe the study is interesting and innovative, but here I suggest some points the Authors could find interesting to discuss on:

Authors’ reply: Thank you for the very positive and constructive feedback! We sincerely appreciate the time this reviewer took to provide a detailed feedback on specific points throughout the manuscript.

Reviewer’s comment: The authors state in the abstract and throughout the text that they performed a combined metabolomics and proteomics analysis, but, in my opinion, the metabolomics and proteomics data are not integrated. Therefore it is advisable to try to integrate the omics data or otherwise remove the word "combined";

Authors’ reply: Thank you for this point. As we do agree with the reviewer, we decided to perform a combined metabolomics and proteomics pathway analysis to highlight the multi-omics-determined pathways that are enriched in the positive and negative correlates to pRBC transfusion, respectively. This is now Figure 8 of the paper.

Reviewer’s comment: In the Results section 2.1, as well as in the abstract, 95 patients are cited, but counting the patients of the four subgroups the patients seem to be 108, the casuistry used should be better explained;

Authors’ reply: We apologize for the issue. The initial studies from the COMBAT project involved 95 subjects (https://www.ncbi.nlm.nih.gov/pmc/articles/PMC5573618/). However, enrolling continued for another year, and subsequent studies were expanded to 118 subjects (https://pubmed.ncbi.nlm.nih.gov/35610738/). We now refer to 118 initial subjects (only 61 receiving transfusion) through the whole paper.

Reviewer’s comment: Figure 1F is described in the same section, but it is not located between the panels of Figure 1;

Authors’ reply: Thanks for catching that. Initially, we had split panels for male and female subjects, but we ended up merging them via overlap in panel E. As such, Figure 1.F should be 1.E in the paper. This has now been corrected throughout. Thanks for catching that and sorry for the inconvenience!

Reviewer’s comment: The temporal trends present in Figures 2, 5, and 6 are difficult to read as the shades of blue colors are difficult to correlate to the reference pRBC Units; I recommend using 5 more diversified colors;

Authors’ reply: All the line plots were revised. A standard blue color palette from very light to very dark was used for the revised plots. We do believe that the revised plots look much better, and thank the reviewer for the suggestion.

Reviewer’s comment: The description in Figure 1 seems very close to a paragraph of results; the caption should be reformulated by making only the description of the panels in the figure;

Authors’ reply: The legend has been revised following this reviewer’s suggestion.

Reviewer’s comment: In 2.2 Paragraph Figure 1 is cited instead to Figure 2, please check carefully;

Authors’ reply: Revised as noted in the two instances this occurred. Thank you for noticing.

Reviewer’s comment: The discussion of the data seems to be very unbalanced in the context of metabolomics and not very thorough with respect to the data obtained from proteomics. An enrichment analysis or a  pathways analysis of proteomics data or integrated ones (metabolites and proteins) may help the reading the obtained results in a functional point of view.

Authors’ reply: This combined pathway analysis has been performed and now presented as figure 8. A discussion of this figure has now been included, with a focus on the proteome.

Reviewer’s comment: In the Introduction (lines 97-99) the sentence could be simplified for greater clarity in the reading phase.

Authors’ reply: The sentence has been reworded.

Reviewer’s comment:   In the captions of the figures, punctuation could be used to separate the description of each panel that make up the figure (example: “A) pRBC transfusions […]” - line 154).

Authors’ reply: Figure legends were revised accordingly.

Reviewer’s comment: In the Results section 2.3 there is probably a typo in the phrase “This group included (decreases in) circulating markers of […]” - line 190.

Authors’ reply: Revised for clarity.

Reviewer’s comment: In the Figure 6 of the Results section 2.4 probably the point is missing from the generic description of the figure (line 253) and in line 258 probably the authors wanted to indicate “protein” instead of “metabolite”.

Authors’ reply: Revised accordingly. Thank you!

Reviewer’s comment: In the Figure 7 of the same section there is probably a typo in the phrase “Box plots of select proteins […]” - line 268.

Authors’ reply: The description of that panel was revised.

Reviewer’s comment:    In the Discussion section there is probably a typo in the phrase “was associated with decreases the circulating […]” - line 290 - and at line 294 the word “and” is missing before “lactate”.

Authors’ reply: Revised accordingly.

Reviewer’s comment:   In the same section al line 327 maybe the “,” should be replaced with “-“ as in the previous line 326.

Authors’ reply: Revised accordingly.

Reviewer’s comment:       In the Materials and Methods section 4.2 gaps may be left as in the following example: SBP<70mmHg - line 364, 369.

Authors’ reply: Revised accordingly.

Reviewer’s comment:        In the Materials and Methods section 4.4 the authors could double check the syntax of the sentence at line 385 and the unit of measure described at lines 401, 405, 425. Moreover, there are some extra spaces at lines 388, 406, 411. At the end, the point is missing at the end of the sentence at line 426.

Authors’ reply: All the units mentioned in these lines were double checked for accuracy.

Reviewer’s comment:       In the Materials and Methods section 4.5 the words “pre”, “post” and “during” should be written in quotation marks to facilitate reading.

Authors’ reply: Revised accordingly.

Reviewer 3 Report

Dear Authors

Topic about roles of erythrocytes in human health and disease 2.0 is very

interesting. I have no comments.

Author Response

REVIEWER 3  Dear Authors, Topic about roles of erythrocytes in human health and disease 2.0 is very interesting. I have no comments.

Authors’ reply: Thank you for the enthusiastic feedback!

Round 2

Reviewer 2 Report

I hope my comments were helpful, the work is definitely suitable for publication.